# Experiences of Changes in Eating Habits and Eating Behaviors of Women First Diagnosed with Gestational Diabetes

**DOI:** 10.3390/ijerph18168774

**Published:** 2021-08-19

**Authors:** Hye-Jin Kim, Eunjeong Cho, Gisoo Shin

**Affiliations:** 1Department of Nursing, Changwon Moonsung University, 91 Chunghonro, Seongsan-gu, Changwon 51410, Korea; 325khj@hanmail.net; 2College of Nursing, Chung-Ang University, 84 Dongjak-gu, Heukseok-ro, Seoul 06974, Korea; foryou0913@cau.ac.kr

**Keywords:** women, gestational diabetes mellitus, eating habit, eating behavior, theme analysis

## Abstract

As gestational diabetes, which is increasing steadily around the world, can cause complications in the mother and fetus, it is essential to change eating habits and eating behavior to prevent this. According to the 2020 American Diabetes Association recommendations, the food plan should be designed for the adequate calorie intake to achieve glycemic goals and consequently promote maternal and fetal health. Thus, the following study has used the qualitative theme analysis method to assess what it means for 28 South Korean women, who were diagnosed with gestational diabetes for the first time, to change their eating habits and behaviors. As a result, themes were derived related to reflection on daily life, formation of new relationships in the same group, efforts that must be made, rediscovery of couples, and lifestyles reborn as new roles. Based on the results of the study, it is shown that the study participants recovered the peace in their mental state after the crisis of gestational diabetes to pursue relaxation and ultimately higher quality of life by following the plan to fulfill healthy achievements, such as changing their eating habits and behaviors. Therefore, future research and support measures to help the healthy behaviors should be sought by comprehensively exploring the effects of women’s experiences in changing their eating habits and behaviors.

## 1. Introduction

Gestational diabetes mellitus (hereinafter GDM) is defined as the first detection of hyperglycemia during pregnancy, and it is diagnosed between 24 and 28 weeks of pregnancy [1,2,3]. With its prevalence clearly growing worldwide—specifically in South Korea, it is also rising—the trend has been suggested to be due to an inflation in the average maternal age at childbirth and a problem resulting from a lack of exercise [4]. Although the risk factors for GDM are similar to those for type 2 diabetes, its onset is known to be affected by overweightness, lack of exercise, and environmental factors and eating habits [5]. In addition, GDM has been reported to cause various physiological negative effects in pregnancy and on the fetus such as prematurity and fetal macrosomia [3]. In addition, diabetic symptoms appear during pregnancy in women who have not previously been diagnosed with diabetes, and the GDM affects not only the mother but also the fetus, which causes stress such as feelings of guilt and responsibility [6].

Therefore, to prevent complications of mother and fetus due to GDM, it is recommended to perform at least 150 min of moderately-intense physical activity each week unless exercising is impossible due to obstetric problems [7]. In addition, it is necessary to receive nutrition education by a diabetes educator; the purpose of nutrition education is to provide energy for the mother and fetus, control blood sugar, and prevent ketosis [8]. In addition, women diagnosed with GDM try to limit carbohydrate intake first in terms of their eating habits, but at least 175 g of carbohydrates must be consumed per day. This is because it provides glucose to the fetus’ brain and is also used in the mother’s energy metabolism [9]. However, it is recommended to consume complex sugars rather than simple sugars, and in addition, by distributing three meals a day and two to three snacks a day, the increase in postprandial hyperglycemia due to carbohydrate intake should be reduced [8].

Eating habits start from infancy and are formed over a long period of time, and eating habits that have already been formed do not change easily and tend to persist until old age [10]. Except for simply maintaining and promoting physical health, eating food affects psychological health such as mood and emotions. Therefore, eating habits form a very close relationship with an individual’s way of life, including social and cultural meaning [11]. Sometimes, eating habits reflect an individual’s inner personality and may satisfy their inner individual needs, and in some cases, even if you are not hungry, food is eaten to reduce loneliness or isolation, or to pursue a sense of social belonging or happiness [12]. In a similar context, there is something called comfort foods, and comfort foods are foods whose consumption evokes a psychologically comfortable and pleasurable state for a person [13]. These comfort food preferences are generally known to be influenced by an individual’s social and emotional experiences, and in particular, previous studies have reported that childhood experiences are important for the formation of lifelong eating habits and eating behaviors, including comfort foods [14].

Eating behaviors refer to the psychological process that occurs due to the intake of food, and the results of these behaviors are formed by the total amount of food consumed during the day or the total amount of nutrients, the time or duration of food consumption during the day, the rate of food intake, and various factors involved in food intake [15]. Eating behaviors basically consist of reactions to sensory stimuli, but these reactions are also mediated by thoughts and emotions [16].

Pregnancy in women is accompanied by the mixed ambivalence of joy and anxiety, because it is the new life task that women have never experienced before. However, women also undergo stress as they must adapt to the new physical discomforts that arise from pregnancy [17]. Furthermore, due to the hormonal changes, their eating habits and behaviors may change during pregnancy [18]. Especially, the women diagnosed with GDM often develop new concerns on consuming the foods they once used to eat freely without restrictions. They may be suddenly required to stop eating the comfort foods they previously enjoyed and change their eating habits and behaviors, leading to difficulties and confusion about the food intake [19].

‘Taegyo’, the Korean traditional practices of prenatal education, recommends what foods should be consumed during pregnancy [20]. The tradition not only encourages pregnant women to choose their foods based on the food’s geometrical shape rather than nutritional quality; it is suggested that white rice should be eaten three times a day [21].

However, according to the results of a recent study on the diet of Koreans, carbohydrates accounted for the highest proportion among carbohydrates, proteins, and fats, and it has also been reported that the risk of carbohydrate addiction, which becomes depressed by not eating carbohydrates, is also high [22]. These Koreans’ eating habits are suggested as one of the main influencing factors of obesity and diabetes, which are gradually increasing [23], and moreover, as food intake broadcasts become popular through social network service (SNS), the risk is increasing more [24]. Therefore, women who were exposed to the cultural and social environment of Korea without any restrictions until pregnancy, after being diagnosed with GDM, experience changes in eating habits and eating behaviors that were not previously predicted [25].

Meanwhile, Ajzen’s theory [26] of planned behavior explains human behavior through variables of attitude, subjective norm, perceived behavioral control, and behavioral intention related to behavior. The main variable that determines an individual’s behavior in the theory of planned behavior is behavioral intention, and the perceived behavioral control directly affects behavioral intention.

Thus, in this study, a semi-structured questionnaire was completed based on the theory of planned behavior for women who were diagnosed with GDM for the first time, and based on this, we attempted to in-depth grasp the meaning of their eating habits and changes in eating behaviors. The results of this study are intended to be used as basic data to help their health behavior in the future by understanding women who must adapt to changes in their eating habits and eating behaviors.

## 2. Methods

### 2.1. Research Design

This study is a qualitative study that analyzed data using a qualitative thematic analysis method [27] to explore the changes and meanings of eating habits and eating behaviors experienced by women who were diagnosed with GDM for the first time.

### 2.2. Participants

The present study included 28 pregnant women who had been diagnosed with GDM in South Korea. After listening to an explanation of the in-depth interview, they completed the application form and the interview agreement.

### 2.3. Data Collection

In-depth interviews were conducted with research participants from January to June 2018 to collect data on the eating habits and eating behaviors of women who were diagnosed with GDM for the first time. In-depth interviews were conducted with the participants who voluntarily agreed to participate in the study in writing. The interview was recorded and then a transcription process was performed. Interviews were conducted using semi-structured questions and specific interview questions as follows. These were based on the theory of planned behavior [26]; researchers created the questionnaire, and the validity of the content was verified by experts in theme analysis and qualitative research (Figure 1).

The locations for interview were chosen by the study participants, so they could participate in the interview in a comfortable environment. Each participant went through one to three rounds of interviews with the semi-structured questions until nothing new was revealed in their further answers. The average length of all the participants’ interviews was 90 min long.

### 2.4. Data Analysis

This study used an inductive thematic analysis method [27] to better understand the changes in eating habits and eating behaviors of women diagnosed with GDM. Thematic analysis is a method used to derive a central theme by finding a concept that is an object of description from data collected through interviews, observations, and field memos, and grasping the relationship between concepts. Significant words, sentences, and paragraphs were primarily open-coded from the collected data and then classified into similar sentences and paragraphs, and the classified sentences and paragraphs were categorized and named. In addition, to secure the reliability and validity of the research, the research was conducted according to the evaluation criteria of qualitative research [28,29]. To secure factual value, after the in-depth interview was over, the researcher summarized the interview details to the participants, confirming that the interview contents were not different from the facts, and there was a procedure for showing the results of the analysis to the participants and confirming their meaning.

### 2.5. Ethical Considerations

This study was conducted after obtaining approval from the IRB of University in South Korea (1041078-201705-HRSB-097-01). After providing a sufficient explanation of the purpose of the study, confidentiality of the data, and disposal of the data after completing the study, we received the written consent of voluntary participation in the study. While participating in this study, it was explained that the subject can choose to discontinue at any time during the study, and that there were no disadvantages after the study participation was withdrawn. In addition, it was explained that the collected research data are not used for any purpose other than research, and they are discarded after data storage by the Bioethics Act.

## 3. Results

### 3.1. General Characteristics of Research Participants

The mean age of the participants was 35.9 years, with ages ranging between 28 years and 44 years. The participants were in the third trimester diagnosed with GDM at 24 weeks of gestation or at 28 weeks of gestation, and there were 19 primiparas and 9 multiparas. It was found that nine participants had a family history of diabetes, and that most participants did not have any pregnancy smoking but did participate in drinking. While the results were analyzed and presented as themes, each theme presents the testimonials of pregnant women who were diagnosed with GDM (Table 1).

### 3.2. Theme 1: Reflection on Daily Life

The first theme, ‘Reflection on daily life,’ is the experiences of the participants immediately after they were first diagnosed with GDM. Participants of the study said that after being diagnosed with GDM for the first time, they doubted the diagnosis itself or that the diagnosis of GDM came in shock. Afterwards, the most frequently mentioned and emphasized by the participants was to look back on their past. Looking back on the past was related to reflection on habits and behaviors that had been unintentionally performed in life. Additionally, they tried to analyze carefully what was the problem of eating habits and eating behaviors, including what kinds of food they ate in the past, how much food they ate for one meal, and what was eaten for midnight snacks and how much. In addition, they had time to reflect on what foods they ate because of pregnancy they had not eaten before and what foods their parents gave them as prenatal foods. In addition, they attempted to check whether these eating habits and eating behaviors were the cause of GDM.

When I was diagnosed with “GDM”, I cried badly alone. I never thought I’d get “GDM”. I literally see the figures with the blood sugar test machine. I do not feel anything in my body. “This” situation “does” not feel realistic.

If you are pregnant, they say that you “should” not move, you have to eat well... If you want to “birth” a pretty daughter, eat a lot of pretty fruit... Unconditionally I think I had a lot of apple juice and grape juice.

My husband and I enjoyed eating freshly baked croissants every morning to our heart’s content. After I was pregnant, they said that I had to eat better than before, so it seems that I only ate various kinds of bread for three meals a day. Also various pies and fancy cakes... I remember all the foods I ate and enjoyed last time.

### 3.3. Theme 2: Create a New Relationship with the Same Group

The second theme, ‘creating a new relationship with the same group,’ began in the process of obtaining information on GDM as well as reflecting on daily life by the research participants. All of the research participants searched through the internet, where necessary information can be easily accessed, and through this process, it was found that women who had been diagnosed with GDM in the past or who were currently diagnosed with GDM joined social media meetings without hesitation. Above all, the participants had a strong sense of confidence in finding and acquiring information from the actual experiences of pregnant women diagnosed with GDM before them. They had no social relationship in the past, but now that they had the commonality of GDM, they were empathizing with each other, understanding and encouraging each other, and forming new social relationships voluntarily. Sometimes through social media, they communicated more actively than with other people with whom they had formed and maintained social relationships, and they tried to maintain a strong sense of bond with common interests. Moreover, they said that they were able to feel relief and comfort by expressing their worries and anxieties that they could not even tell their family on social media.

I spent all day searching for information on the Internet SNS. The people I met at the “SNS” meeting are the people I had never met and didn’t even know their faces in the past, but I was able to get close to them right away. Because of gestational diabetes, we could sympathize and understand each other.

The food recipes or foods I shouldn’t eat and the frustrations and despairs that those who experienced “GDM” experienced told me at the social media gatherings served as a compass for what I ate and how to do.

If I had more time chatting with friends on the phone before, now I have more time to spend at “SNS” meetings. When I read the articles posted on “SNS”, it was so comforting.

### 3.4. Theme 3: Efforts That Must Be Made

The third theme, ‘Efforts that must be made,’ was made in the process of conflict that the research participants faced while implementing changes in their eating habits and eating behaviors. This is because the participants had difficulties in restraining their free eating habits and enjoyment of eating behavior without any existing restrictions. Moreover, the participants who lived at work or received invitations to eat from acquaintances experienced that they were influenced by an environment in which it was difficult to control and adjust eating habits and eating behaviors only with their own will and effort. In addition, the participants were uncomfortable and stressed about the health behavior of measuring blood sugar at a set time every day and recording what they ate each day. On the one hand, they also experienced a feeling of psychological burden, such as having nightmares of giving birth to a baby that is not normal. However, among these conflicts, discomfort, and burden, the participants tried to control their own eating habits and eating behaviors for the health of the fetus, and furthermore, strong fetal attachment was expressed through movement of the fetus in the stomach. All of the research participants tried to achieve changes in eating habits and eating behaviors to achieve the goal of changing eating habits and eating behaviors.

Previously, I had lunch with my coworkers, but now, I eat the packed lunches by myself. I had the pleasure of eating together... At lunchtime, it sometimes becomes a conflict. However, because the foods that can be eaten out are usually strong and stimulating... Because I often eat noodles...

It’s stressful to check my blood sugar at a set time every day. It is also burdensome to inform my colleagues about my situation. But as soon as I feel my baby moving in my stomach, I feel guilty in my heart and I am really sorry for the baby. I shouldn’t think like this, and I reflect on it.

The fruit and ice cream that I had as a dessert after meals... I can’t eat it anymore... I’m upset, but I have to try. No, I mean, I definitely want to have a change in eating habits. Above all, it is for the baby’s health.

### 3.5. Theme 4: Couples Reborn in New Roles

The diagnosis of GDM did not only require the change of the participants: it also meant that the husband, who had lived and became accustomed to their habits, also had to change. Thus, the participants of the study all agreed on the theme of ‘Couples reborn in new roles,’ and it was found that the support of the husband played a role in the participants’ eating habits and dietary changes. In particular, one of the changes of husbands is that their questions became clear and concrete. Before, husbands asked participants vague questions about the well-being of the participant and the fetus, but now they wanted to specifically ask and know what the normal range of blood sugar is, what foods participants should not eat, whether the fetus is growing normally, and what the signs of danger are to the participant and fetus. In addition, husbands also actively participated in changes in lifestyle, such as self-regulation of food such as cakes, fried chicken, beer, and soda, which they previously enjoyed without restrictions, and walking with participants. Such support from husbands not only acts as a factor that strengthens the intimacy of the participants with their husbands, but at the same time, it was positively affecting the acquisition of the role as a parent as well.

My husband helped me a lot. He stopped the fried chicken we ate together for a midnight snack, and didn’t drink beer in front of me... We also squeeze a diabetic diet together... We walk together as soon as we can… My husband’s daily life has also changed.

Before, my husband didn’t ask questions to the doctor when I went to the prenatal examination. But now, whether the baby is growing properly... whether the glucose level is normal... I ate food like this, but is it okay... His interest in the baby has also increased, and his help and caring for me have also grown.

My husband and I thought that when you get pregnant, you will naturally become parents. Both my husband and I came to realize that it takes constant effort to become healthy parents.

### 3.6. Theme 5: Rediscovery of Lifestyle

The fifth theme, ‘Rediscovery of lifestyle,’ was acting as a motive for continuing changes in the eating habits and eating behaviors of the participants. Before being diagnosed with GDM, due to busy daily life similar to most modern people, the usual meal times of the lives of participants were also irregular, and occasionally, there were many cases of eating late. In addition, to eat a meal within a short time, it was common to eat fast food, instant food, delivery food, etc. that can be easily purchased or cooked. On top of that, when exposed to food temptation or stressed through TV or the Internet, they often overate, and their physical activity was greatly reduced. This was also caused by the time constraints of busy daily life, but on the other side, every participant agreed that the cause was also the lack of psychological space for proper eating habits and eating behaviors. Participants reinforced changes in eating habits and eating behaviors to maintain them as they experienced changes in their bodies, such as changes in blood sugar. In addition, they said that they gained psychological relaxation in the process of choosing food ingredients and making and preparing food by themselves for the health of the fetus and their family. Furthermore, this lifestyle transformation process had an impact on the motivation for pursuing health and quality of life.

I used to be busy every day, so I relied on eating fast foods and delivery foods. I could not even think of spending an extra time to cook a meal unless it was for a special anniversary. Now that I look back, I actually did have enough time to invest in preparing the foods for myself and my baby. I was just too mentally busy to afford the thought of spending time on cooking foods.

Since I now buy food ingredients and make meals with my own hands, for the first time, I have realized the values of devoting myself into making good meals and eating well. I used to eat whichever foods that I bought on that day without much thoughts, but my diet choice is now based on the goal to be healthy and live well for a long time.

If I eat healthy, my blood sugar drops and my body feels lighter... and so I keep trying.... Of course, I thought about the health of my baby and my family through food, and I also thought about the quality of life.

## 4. Discussion

This study attempted to find out the significance of the experiences of changing eating habits and eating behaviors of women diagnosed with GDM for the first time. Based on the results derived from this study, we intend to discuss the meaning of changes in eating habits and eating behaviors they experienced since the diagnosis of GDM.

Risk factors for GDM are known to be affected by hormonal changes and genetic factors as well as environmental factors such as obesity, diet, and lack of exercise [1,2,3,5]. Among environmental factors, food has been reported as an important factor, and it has been suggested that the frequent consumption of carbohydrates composed of simple sugars or animal fats is closely related to diabetes [8,9]. According to the research results that investigated the relationship between women with GDM and their eating habits in South Korea, it was reported that pregnant women diagnosed with GDM had irregular meal times, meals, and frequency of meals, and that they enjoyed simple sugar and meat such as rice rather than vegetables, but they also enjoyed midnight meals and snacks [30,31]. Not only that, but as a dessert, eating habits such as enjoying fruit immediately after eating or eating large amounts of fruit such as oranges and apples with juice called ‘juice for health’ were also reported to be associated with diabetes [32]. Participants of this study also experienced a ‘reflection on daily life’ to look back on whether there were problems with eating habits and eating behaviors up to now. As a result of the shock and embarrassing negative feelings for the diagnosis of GDM while undergoing a major change in pregnancy in the life cycle of these women [6], they started to change their eating habits and eating behaviors.

Most of us living in modern society use the Internet and SNS almost every day [33]. Participants in this study also used the Internet and SNS in their daily life. After the diagnosis of GDM, their use of SNS changed to a special meaning of ‘creating a new relationship with the same group.’ Usually, SNS users use SNS for various purposes such as expression of identity, social interaction, usefulness, exchange, participation in information search, and communication [34]. Although the SNS activities of the participants in this study started with the purpose of obtaining information, they experienced that as GDM became common, it developed into emotional intimacy beyond the dimension of information sharing. In the formation of social relationships, intimacy is known to play a key role in the process of mutual self-disclosure [35]. Such self-exposure is more easily achieved in the same group or groups with common elements, and through this, it was found that it plays a positive buffer against stress or negative psychological state experienced by individuals [36]. In addition, it is known that empathy based on commonality, emotional encouragement, or providing advice or information about knowledge through experience provides strong trust and psychological stability [37]. Therefore, it can be seen that the formation of relationships through the SNS activities of the participants in this study alleviates the shock and anxiety about the diagnosis of GDM, and it played a positive role in their eating habits and dietary changes.

However, on the one hand, it was found that the participants were still experiencing conflicts with their daily lives and limitations due to their surrounding environment. All of the participants agreed that the change in eating habits and eating behaviors is a ‘must-do effort’, which is in some aspect true in that there are some areas where changes in eating habits and eating behaviors are formed, but in another aspect, it supports the fact that there is also a conflict about daily life. Eating habits and eating behaviors are usually formed from infancy and are continuously dominated by habits in late adulthood [38]. These reports suggest that even if adult women perceive health risks, adjust their diet, and choose safe foods, it means that it is difficult to change the eating habits and eating behaviors that have already been formed over a long period of time [39]. The results of a study on the nutritional status of pregnant women in South Korea also support these results, reporting that even after pregnancy, pregnant women still had fatty meat-centered meals and wrong eating habits that were salty and excessive, and they enjoy frequent consumption of sweet foods, overeating, and eating out [30,31,32]. Therefore, specialized nutritional knowledge education is needed to prevent high-risk pregnancy such as GDM or gestational hypertension, and many improvements are required in terms of the diversity and balance of food choices [40]. Even among these restrictions, the participants of this study showed a strong sense of responsibility for changes in eating habits and eating behaviors for the health of the fetus, which was the same as the study results indicating that the stress of pregnant women due to GDM did not negatively affect fetal attachment or maternal identity [41].

It was confirmed that the support of spouses was influencing the reinforcement of fetal attachment and changes in the eating habits and eating behaviors of the participants. Spousal support is an act of receiving consideration, help, comfort, and support from a spouse, which is an important factor influencing psychological stability and perceived sense of control in implementing planned health behaviors [42]. The participants said that the marital relationship could be stronger than before, which infers that the behavior of husbands who listened and sympathized so that they could motivate themselves in difficult situations worked. By sharing feelings about the health and safety of the fetus, the experience of the ‘couples reborn in new roles’ was acting as another trigger of attachment between father and fetus.

The reason why the changes in eating habits and eating behaviors of the participants of this study could be maintained continuously from GDM to the third trimester was from the experience of ‘rediscovery of the lifestyle’ which was the materials needed for the changes. The intention for behavioral change shows how much an individual is willing to try and how much effort is made to perform a specific behavior. These behavioral changes can be sustained for a long time under the influence of attitudes, subjective norms, and perceived behavioral control [26]. It is necessary to repeat internal motivation to continuously maintain this behavioral control, which can be achieved by self-confidence and psychological relaxation based on the experience of controlling behavior [43]. These factors ultimately act as prerequisites in pursuing health promotion and quality of life.

This study has the following limitations. First, as the results of this study are the results of in-depth interviews with women with GDM in South Korea, there is a limit to generalization on behalf of all GDM women. Second, in analyzing the results of this study, as questions that can grasp the relationship between changes in eating habits and eating behaviors taking into account the general characteristics of participants, such as age, educational attainment, and diabetic family history, are not addressed, there is a limitation in not being able to grasp the context in detail. Lastly, from the results of this study, since the participants who experienced positive eating habits and dietary changes participated as a sample, there is a limitation in not being able to derive research results that reflect the frustrations or negative experiences of eating habits and dietary changes. Therefore, based on the results of this study, it is necessary in the future to compare and analyze the eating habits and dietary changes of women with GDM by applying various conditions such as national, regional, and individual characteristics in depth.

## 5. Conclusions

This study attempted to identify the changes in eating habits and eating behaviors and their meaning for women who were diagnosed with GDM for the first time in South Korea using the thematic analysis method [27]. As a result, the five themes of reflection on daily life, creating a new relationship with the same group, efforts that must be made, couples reborn in new roles, and rediscovery of lifestyle have been identified. After the diagnosis of GDM, women had time to reflect on their daily lives, and at the same time, they formed a new relationship with other female GDM patients through SNS under the common interest of GDM. Although they experienced conflicts within oneself due to the habits and circumstances that retained through their life, they made strong efforts to control their own behaviors for the health of the fetus. The spouse’s support had a positive effect on their efforts, and by rediscovering their lifestyle through change, the participants in the study continued to achieve internal synchronization with changes in eating habits and eating behaviors. In addition, changes in the eating habits and eating behaviors of the participants in this study became a medium that could transition from the crisis of GDM to planned health behaviors, and through these experiences, it can be seen that this experience acts as an internal motive in recovering psychological relaxation and ultimately promoting health and pursuing quality of life. Therefore, in the future, by comprehensively exploring the experiences of changing eating habits and eating behaviors in times of unexpected crisis in the life cycle of women, research strategies that can help their health behaviors should be sought.

## Figures and Tables

**Figure 1 ijerph-18-08774-f001:**
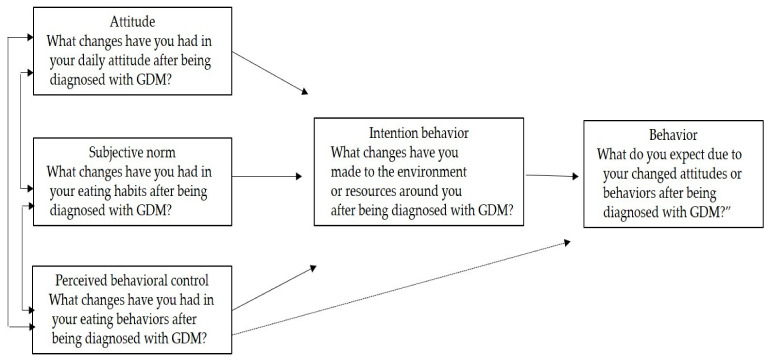
The flowchart showing the questions based on the theory of planned behavior.

**Table 1 ijerph-18-08774-t001:** The characteristics of the study participants.

	Age (Years)	Gravidity and Parity	Employed/Housewife	Has Family History of DM
1	35	G1P0	Employed	Yes
2	28	G1P0	Housewife	None
3	28	G1P0	Employed	Yes
4	42	G1P0	Housewife	None
5	37	G1P0	Employed	None
6	33	G2P1	Employed	Yes
7	35	G1P0	Housewife	None
8	35	G2P1	Employed	Yes
9	40	G1P0	Employed	None
10	31	G2P1	Housewife	None
11	42	G1P0	Housewife	None
12	43	G2P1	Employed	Yes
13	42	G2P1	Employed	None
14	36	G1P0	Employed	None
15	40	G1P0	Employed	Yes
16	35	G2P1	Employed	None
17	35	G1P0	Housewife	None
18	34	G2P1	Employed	Yes
19	35	G1P0	Employed	None
20	39	G1P0	Housewife	None
21	36	G1P0	Employed	Yes
22	36	G1P0	Employed	None
23	32	G2P1	Housewife	None
24	34	G1P0	Employed	None
25	44	G1P0	Employed	Yes
26	39	G1P0	Housewife	Yes
27	29	G2P1	Housewife	None
28	30	G1P0	Employed	Yes

## Data Availability

The data presented in this study are available on request from the corresponding author.

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
