# Peer review of "Experiences of Changes in Eating Habits and Eating Behaviors of Women First Diagnosed with Gestational Diabetes"

_ijerph, 2021, doi:10.3390/ijerph18168774_

Round 1

Reviewer 1 Report

The manuscript is on an important topic and is well-written. Most of my comments are on editing for English. I have one major suggestion however – the significance of your findings and the practical applications are not very clear. Could you please provide some specific guidance for women who have been diagnosed with gestational diabetes in your abstract and in the conclusions?

Line 36: Change “has not” to “have not”

Line 42: With regards to recommendations for physical activity for pregnant women, it might be best to refer to the latest guidelines (please see Mottola MF, et al. Br J Sports Med 2018;52:1339–1346. doi:10.1136/bjsports-2018-100056).

Lines 71-72: Sentence starting with “Most of all…” – please re-word this sentence for clarity

Line 81: Change “culture” to “cultures” and delete the word “that”

Line 82: At the end of this line, change “pregnancy women” to “pregnant women”

Line 83: Change “shape food” to “shape of food”

Line 99: Change “written” to “completed”

Line 102: Change “These results” to “The results”

Line 120: Delete the word “are”

Figure 1: Is this figure necessary, given that the same questions are described in the text? I suggest either deleting the figure or the text to reduce redundancy.

Page 6, line 9 (please note that the line numbering was reset to 1 on this page): Delete the word “once” at the end of this line.

Table 1: I don’t think this table is necessary since all of this has been summarized in the text. I suggest deleting.

Line 156: Change “eating” to “ate”

Lines 168-169: Sentence starting with “Looking back…” – Please re-word this sentence.

Reviewer 2 Report

The study in its present form has many limitations and I strongly suggest major revisions before resubmitting. I will state my comments below:

1) Please consider modifying table 1 with mean values for age, percent of patients with a family history of DM, and percent of patients employed. Also, if available, mean weight +- standard deviation (if values are normally distributed)

2) How were the data analyzed? There is no section with the results. 

3) The conclusion is very general, has no correlation with the provided data, the results are missing.

4) The included study group is very small to generate an important scientific value

Reviewer 3 Report

Manuscript: Experiences of Changes in Eating Habits and Eating Behaviors of Women First Diagnosed with Gestational Diabetes

In this study on changes in eating habits of women with gestational diabetes, the authors have done a good job gathering information. The manuscript is well structured and I consider that its strategy contributes as an alternative to promote the commitment of the patients to achieve the metabolic goals and thus minimize complications both for them and for the fetus. I believe that the manuscript can be published if improvements are made in the writing and the English.

  1. Keywords: Typo error, please correct from “Thema analysis” to “Theme Analysis”.
  2. Page 2, Lines 71-80: Please proofread paragraph, improve structure.
  3. Page 2, Line 71: Please consider substituting “Most of all” for “Especially”
  4. Page 2, Line 81; Line 82: Please rewrite sentences.
  5. Page 2, Line 90: (SNS) please spell in full the word, then abbreviate in parenthesis when used for the first time.
  6. Page 3, Line 21: Consider using “questionnaire” instead of “form of questions”
  7. Page 5, Line 10: Could you please explain “the number of interviews was from 1 to 3 times.” How many interviews were patients supposed to complete? Were they all able to complete them? Were the same questions asked in each interview? If these were follow-up interviews, did you observe any changes?
  8. In the results section, I suggest you add a sentence indicating that results were analyzed and presented by theme and that each theme also presents patient’s testimonials.
  9. Page 9, Line 189: Please proofread sentence.
  10. Page 9, Line 199: Consider substituting “as” for “in”.
  11. Page 10, Line 222: Please substitute “later adulthood” for “late adulthood”.
  12. It would have been interesting to observe how this intervention impacted the metabolic outcomes of these patients. (for example HbA1c%).

Round 2

Reviewer 1 Report

The authors have addressed all my comments

Reviewer 2 Report

The authors have revised the text as suggested.